# The process of attrition in pre-medical studies: A large-scale analysis across 102 schools

**Charlene Zhang** *, **Nathan R. Kuncel, Paul R. Sackett**

University of Minnesota, Minneapolis, Minnesota, United States of America

* zhan5449@umn.edu

**Data Availability Statement:** The dataset used for this project is proprietary, owned and collected by a third-party, and cannot be freely posted online. The authors did not have any special access to the

## Abstract

The important but difficult choice of vocational trajectory often takes place in college, beginning with majoring in a subject and taking relevant coursework. Of all possible disciplines, pre-medical studies are often not a formally defined major but pursued by a substantial proportion of the college population. Understanding students' experiences with pre-med coursework is valuable and understudied, as most research on medical education focuses on the later medical school and residency. We examined the pattern and predictors of attrition at various milestones along the pre-med coursework track during college. Using a College Board dataset, we analyzed a sample of 15,442 students spanning 102 institutions who began their post-secondary education in years between 2006 and 2009. We examined whether students fulfilled the required coursework to remain eligible for medical schools at several milestones: 1) one semester of general chemistry, biology, physics, 2) two semesters of general chemistry, biology, physics, 3) one semester of organic chemistry, and 4) either the second semester of organic chemistry or one semester of biochemistry, and predictors of persistence at each milestone. Only 16.5% of students who intended to major in pre-med graduate college with the required coursework for medical schools. Attrition rates are highest initially but drop as students take more advanced courses. Predictors of persistence include academic preparedness before college (e.g., SAT scores, high school GPA) and college performance (e.g., grades in pre-med courses). Students who perform better academically both in high school and in college courses are more likely to remain eligible for medical school.

## Introduction

All students inevitably face the challenge of choosing their vocational path. For many, the process begins with choosing their college major. This is a difficult but extremely important choice with lasting consequences. Some of the most common regrets of Americans involve their educational and career choices [1]. The present study investigates a particular case of career planning—the process through which undergraduate students fulfill prerequisite coursework for medical school. It is no secret that a substantial proportion of high school graduates aspire to a

data that other researchers would not have. Researchers interested in the data for replication purposes only. May request access by contacting the College Board here The College Board 250 Vesey Street New York, NY 10281 212-713-8088 Or through the direct online portal here: http://research.collegeboard.org/data/request Please reference the paper and authors in the request.

**Funding:** This research was supported by a grant from the College Board to Paul R. Sackett and Nathan R. Kuncel. Paul R. Sackett served as a consultant to the College Board. This relationship has been reviewed and managed by the University of Minnesota in accordance with its conflict of interest policies. This research is derived from data provided by the College Board. Copyright 2006–2011 The College Board. www.collegeboard.com The funders had no role in study design, data collection and analysis, decision to publish, or preparation of the manuscript.

**Competing interests:** The authors have declared that no competing interests exist.

career in the medical field. In fact, health professions and related programs were found to be the second most popular field of study among four-year college students in 2018 [2]. However, this group of hopeful young adults often change their mind at one point or another throughout their education [3,4]. The journey of pursuit in a medical career involves years of strenuous schooling, not to mention the competitive nature of each step along the trajectory. In 2019, applicants to medical schools submitted a median number of 15 applications and only 42.6% of applicants were accepted to any medical program [5,6]. Given the lengthy and strenuous journey of pursuit in a medical career involving years of medical school and residency later and the competitive nature of the field, early years of pre-medical studies in undergraduate is no doubt important. The present study investigates the process through which undergraduate students fulfill prerequisite coursework for medical school. Specifically, we examine rates of attrition at different points in the pre-med curriculum and predictors of continual persistence.

Williams [7] describes the "Pre-Med Syndrome" as the first phase of attrition in the medical education pipeline. Past research has found that upon preparing for medical school application, students are faced with difficult coursework, shattering of unrealistic expectations about what a medical career entails, and harsh admission criteria including high score requirements for the Medical College Assessment Test (MCAT) and college grade point average (GPA). Consistent with this list of challenges, college students report declines in their interest in pre-medical studies [3]. Overall, several groups of factors have been linked to attrition in pre-medical studies, namely demographics, academic preparedness, and coursework. We review each of these below.

Certain demographic characteristics have been associated with lower likelihood of persistence in pre-medical studies, specifically being female, and being a member of underrepresented racial and ethnic minority groups (URM). In a longitudinal study following several cohorts of college freshmen in a single college who indicated interest in pre-medical studies [3], women reported a larger decline in interest in continuing medical education than men, as well as a lower likelihood of applying to medical schools than men. Similar patterns were observed for URM students as compared with non-URM students. However, in another study examining students across six California colleges [8], URM students were found to be almost equal in their likelihood to complete courses required to apply to medical school as non-URM students. On the other hand, demographic characteristics such as having family members who were doctors and higher family income have been found to aid with persistence [3].

Studies have examined whether the gender discrepancy in persistence is coupled with college performance. Fiorentine reported that while males and females with high levels of college performance were equally likely to apply to medical schools, females with low performance were less likely to apply than males with similarly low performance [9]. A normative alternatives explanation was proposed such that differences in gender norms provide males with more disincentives to changing their career trajectories when faced with setbacks [10]. Consistent with this, Lovecchio and Dundes reported gender to be a moderator for the relationship between performance in organic chemistry courses, with women more likely to alter their career plans as a result of poor performance than men [11]. Similarly, pre-medical females construed their own low performance as a sign of poor fit with the medical education track, but not males [12].

The issue of underrepresentation of students from URM groups in medicine and even the broader science, technology, engineering, and mathematics (STEM) field is systemic, beginning with being less prepared in pre-college and college courses [13]. Less access to and participation in AP science courses, lack of support and guidance from family and faculty mentors, and financial challenges can all affect whether students from URM persist [14–16].

Additionally, there has been some suggestion that academic preparedness and performance may influence decisions about persistence in pre-medical education. Among factors prior to college, high school GPA was found to significantly predict pre-medical student retention [17]. However, Barr and colleagues found no association between SAT scores and persisting interest in pre-medical education [3]. In a study by Lovecchio and Dundes, 68% of the former pre-med students surveyed pointed to low grades during college as a major concern for their dropping out [11].

When inquired about specific coursework that deterred students from persisting in their interest in medicine in college, frequently mentioned were low grades obtained in difficult pre-medical "gateway" courses, especially the notorious organic chemistry [3,11,18,19]. Further, the discouraging effects of such chemistry course have been found to be especially pronounced for students from URM groups and women [3,18,20].

While this body of research points to demographics, scholastic preparedness, and college performance as predictors of attrition in the pre-medical curriculum, many causes have been derived qualitatively using small-sample interviews and case studies [3,11,12,18,19,21]. Among studies that quantitatively and longitudinally examined predictors of medical education attrition, many used multiple cohorts from a single institution [22]. A factor that perhaps can partially account for this lack of large-scale, quantitative research on the undergraduate experience of premed students is that pre-medical studies is not a well-defined major in most post-secondary institutions in the United States. Students who are on the pre-med track often major in biological sciences, physical sciences, health sciences, and some even in humanities and social sciences [23]. Therefore, it can be difficult to identify students who are in various majors but are in actuality on the premed track. The present study takes an indirect approach. Rather than attempting to directly identifying the group of pre-med undergraduates, we use all four year of students' coursework data to distinguish those who do not graduate with the basic pre-requisite coursework for medical school from those who do. The combination of this coursework criterion and students' self-reported intentions to pursue a pre-med track before let us reasonably estimate the group of pre-med students.

Furthermore, much of the previous work focused on the singular, final status of persisted versus dropped-out. Thus, the approximately four-year process of pre-medical coursework has been glossed over. The 2019 Matriculating Student Questionnaire (MSQ) administered by the Association of American Medical Colleges (AAMC) reported that while a majority of respondents decided that they wanted to study medicine before college, a substantial percentage (34.8%) decided during their four years of college, most of whom (22.1%) decided during the first two years of college [5]. There is much to be gained from examining the patterns of attrition throughout the various stages or milestones of achieving a pre-medical degree and completing medical school prerequisite courses.

Our paper focuses on progress through the science prerequisites for medical school among students stating that pre-med is their intent when they take the SAT. Using data collected by the College Board, we have a sample of 15,442 students from 102 post-secondary institutions across the United States for whom we have a complete record of course-taking in college. Based on the required courses for entry into medical school, we are able to examine which students remain medical school-eligible at various milestones: 1) one semester of general chemistry, biology, and physics, 2) two semesters of general chemistry, biology, and physics, 3) one semester of organic chemistry, and 4) either the second semester of organic chemistry or one semester of biochemistry, as well as predictors of fulfillment at each milestone.

We note that our focus here is on whether or not a student stating an initial pre-med intent completes the academic requirements to be eligible to apply to medical school, not on whether the student does or does not apply to medical school. This is a consequence of using a large

dataset which contain rich details on individual course-taking provided by 102 colleges and universities. The data are anonymized, prohibiting inquiry into students plans and choices following graduation. This is clearly a limitation of the study, but we view the access to this rare large-scale data base on medical school eligibility as a worthwhile tradeoff.

## Method

### Sample

Data from students who began their post-secondary education in academic years between 2006 and 2009 were provided by The College Board. Of the 917,459 individuals whose intended major choice information was available, 170,866 individuals had four years of complete coursework data available and attended a school using a standard semester system. Of those, 153,512 students did not indicate any intention in studying pre-medicine at the time of SAT, 1,912 indicated pre-medical studies as their secondary or tertiary major choice, or indicated pre-medical studies as their first choice major but were not certain of their choice, and 15,442 indicated pre-medicine as their first choice major and were very or fairly certain of their choice. This resulted in our primary sample of 15,442 students spanning 102 institutions.

### Measures

**Demographics.** Gender and race/ethnicity information were provided. Of 15,442 students, 9,852 (63.80%) were female and 5,590 (36.20%) were male. Other than 331 (2.01%) individuals whose race/ethnicity information was missing, 8,130 (52.65%) were White, 2,899 (11.39%) were Asian or Pacific Islander, 1,764 (11.42%) were Black or African American, 1,674 (10.84%) were Hispanic, 67 (.43%) were American Indian or Alaska native, and 597 (3.87%) identified as Other race/ethnicity.

**Socioeconomic status (SES).** Three SES variables were available: father's education, mother's education, and parental income. Parental income was reported on response options that consisted of several income ranges. A dollar value for parental income is calculated by taking the natural logarithm of the midpoint in each income bracket, thus normalizing the distribution. A equally weighted composite of the three variables were calculated by standardizing each variable individually, summing the three, then standardizing the sum again, following the procedure specified by [24].

**Intended major choice.** Students indicated their intended major choice from a list of 368 different majors at the time they took the SAT.

**High school GPA (hsGPA).** Self-reported GPA was used.

**SAT scores.** Three SAT scores were provided based on the three subsections: Verbal/Critical Reading (SATV), Writing (SATW), and Math (SATM), with possible scores ranging from 200 to 800. A composite SAT score (SATC) was calculated by summing the three subsection scores.

**AP courses.** Data were provided on whether students took any AP courses and their grades on the corresponding AP exams, ranging from 1 to 5. The ones relevant to the pre-medicine curriculum were included: biology, chemistry, calculus AB, calculus BC, english language and composition, english literature and composition, physics B, physics C: electricity and magnetism, physics C: mechanics, and statistics.

**College coursework.** For all college courses that are taken by each student, information was provided about the course name, the year and semester in which the course was taken, the content area in which courses fell, and the grade obtained.

## Procedure

Students' eligibility to apply to medical schools (persistence in studying medicine) was operationalized as whether they fulfill the standard coursework required by most medical programs. To determine medical school prerequisites, we tallied definitions used by prior research [8,9,25], admission requirements specified by the Association of American Medical Colleges (AAMC), and course requirements of 105 medical schools across the United States in 2013.

AAMC suggested general prerequisites to be one year of Biology, one year of Physics, and two years of Chemistry that includes Organic Chemistry courses [26]. Some medical programs also allowed one biochemistry course to substitute for the second organic chemistry course. Therefore, we operationalized fulfillment of medical school prerequisites to include one year (two semesters) of biology, physics, general chemistry, and either one year of organic chemistry or one semester of organic chemistry with one semester of biochemistry.

The number of courses in each subject was counted to indicate continued eligibility for medical school at a number of milestones during coursework progression. For a course to be included, a numerical grade of 0.7 or higher on a scale of 0 to 4 needed to be achieved, as it is the lowest passing grade.

Further, many students receive college credit for achieving satisfactory grades on AP exams. AP exam scores of 3 or above were accepted by most schools as course credits [27], and were counted toward prerequisite fulfillment.

## Analyses

Descriptive statistics were calculated to examine differences between fulfillers of medical school course pre-requisites and non-fulfillers.

Subsequently, logistic regression analyses at several important coursework milestones were performed to examine the effects of demographics, academic preparedness, and college course performance on course fulfillment at each stage.

## Results

### Rates of prerequisite fulfillment

Overall, among the 15,442 students who indicated initial pre-med interests, 2,555 (16.5%) completed the full set of medical school prerequisites, while a small proportion (7.7%) never completed any chemistry, biology, or physics courses. As shown in Fig 1, the process of dropping out is examined at four major milestones, namely:

1. whether students passed one semester of general chemistry, biology, and physics (39% did so);

2. among students who fulfilled milestone 1, whether they passed a second semester of general chemistry, biology, and physics (64% did so);

3. among students who fulfilled milestone 2, whether they passed one semester of organic chemistry (77% did so);

4. among students who fulfilled milestone 3, whether they passed a second semester of organic chemistry or a semester of biochemistry, thereby satisfying the entire fulfillment requirement (85% did so).

Proportions of fulfillers at each milestone were also calculated for each of the 102 institutions individually. Across all institutions, an average of 34% of students fulfilled milestone 1

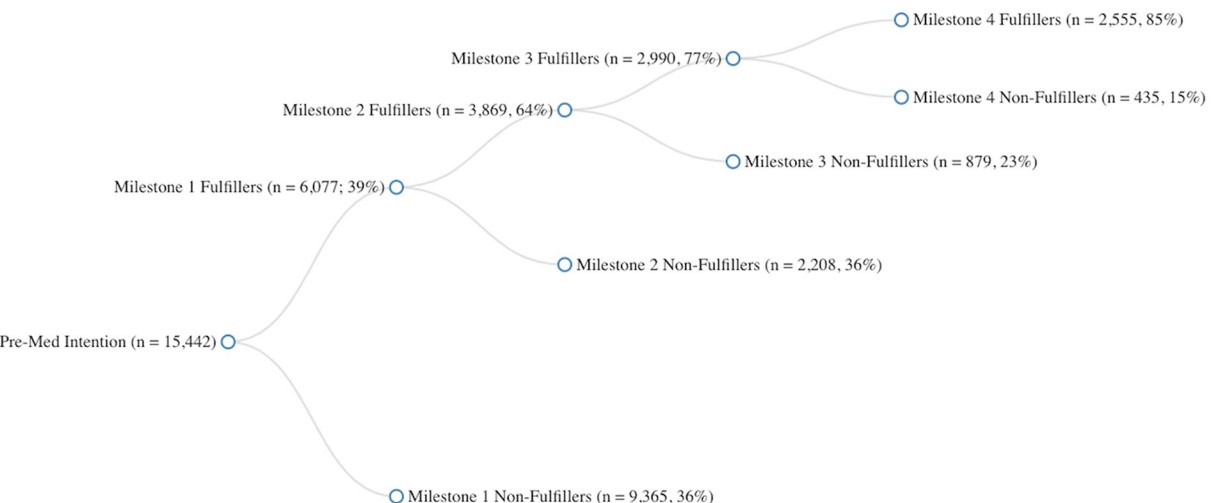

**Fig 1. Pattern of continuing fulfillment of pre-med coursework at various milestones.** Pre-Med Intention = students who indicated intentions of majoring in pre-medicine at the time of taking SAT, Milestone 1 = whether students took a first semester of general chemistry, biology, and physics, Milestone 2 = whether students who completed Milestone1 took a second semester of general chemistry, biology, and physics, Milestone 3 = whether students who completed Milestone2 took a first semester of organic chemistry, Milestone 4 (full fulfillment) = whether students who completed Milestone3 took a second semester of organic chemistry or a semester of biochemistry.

(SD = 19%), 23% fulfilled milestone 2 (SD = 16%), 19% fulfilled milestone 3 (SD = 16%), and 17% fulfilled milestone 4 (SD = 16%).

To allow comparisons, the number of students who fulfilled prerequisites for medical schools was also tallied for the group of 1,912 students who indicated some intention in studying pre-medicine but were not certain, and the group of 153,512 students who had no intention of studying pre-medicine. 267 (14.0%) fulfilled the full set of medical school prerequisite coursework in the former group and 2,633 (3.9%) fulfilled it in the latter group.

## Comparison between fulfillers and non-fulfillers of medical school prerequisites

Demographic information of the 2,555 fulfillers and 12,887 non-fulfillers of medical school prerequisites can be found in Table 1. A larger proportion of males (21%) who reported pre-med intentions fulfilled the prerequisites than females (14%).

With regards to race/ethnicity, rates of fulfilling prerequisites among those who had intention of pursuing pre-medical studies were the highest for Asians (23%), followed by the Other Minority group (20%), then Whites (16%) and Hispanics (13%), and the lowest for Blacks (9%).

There were meaningful differences between fulfillers and non-fulfillers on a variety of variables, both prior to college and throughout college (see Table 2). Fulfillers scored higher than non-fulfillers on the SAT-Combined ($d$ = .44), with the largest difference in the Math section ($d$ = .51). Fulfillers also reported higher GPA, both in high school ($d$ = .33), and in all four years of college (average $d$ = .37). Higher socioeconomic status ($d$ = .18) was reported by fulfillers than non-fulfillers. In cases where AP grades were available, fulfillers obtained higher AP scores than non-fulfillers, including AP Biology ($d$ = .52), AP Chemistry ($d$ = .38), and AP Physics B and C (average $d$ = .50).

When college performance is broken down specifically by course, fulfillers were found to have taken a greater number of medical school prerequisite courses (average $d$ = 1.31) as

**Table 1. Distribution of gender and race of the study population according to prerequisite fulfillers at each milestone: 2006–2009.**

| Group | n | Fulfilled Milestone (%) | | | |
|---|---|---|---|---|---|
| | | M1 | M2 | M3 | M4 |
| Overall | 15,442 | 39.4 | 25.1 | 19.4 | 16.5 |
| Gender | | | | | |
| Male | 5,590 | 45.6 | 30.4 | 23.9 | 21.1 |
| Female | 9,852 | 35.8 | 22.0 | 16.8 | 13.9 |
| Race | | | | | |
| White | 8,130 | 38.2 | 24.2 | 19.1 | 16.3 |
| Asian | 2,899 | 50.8 | 32.6 | 26.1 | 23.3 |
| Black | 1,764 | 30.0 | 17.7 | 12.2 | 9.3 |
| Hispanic | 1,674 | 33.6 | 22.7 | 15.1 | 12.5 |
| American Indian | 67 | 22.4 | 16.4 | 11.9 | 9.0 |
| Other | 597 | 44.4 | 28.5 | 22.9 | 19.6 |
| Race x Gender | | | | | |
| White Male | 3,080 | 43.7 | 29.2 | 23.5 | 20.6 |
| White Female | 5,050 | 34.8 | 21.2 | 16.4 | 13.6 |
| Asian Male | 1,205 | 53.7 | 35.0 | 27.9 | 26.1 |
| Asian Female | 1,694 | 48.8 | 30.9 | 24.8 | 21.2 |
| Black Male | 353 | 39.1 | 24.9 | 17.0 | 12.7 |
| Black Female | 1,411 | 27.8 | 15.9 | 11.1 | 8.5 |
| Hispanic Male | 586 | 41.3 | 30.7 | 21.7 | 18.1 |
| Hispanic Female | 1,088 | 29.5 | 18.4 | 11.6 | 9.6 |
| American Indian Male | 24 | 16.7 | 12.5 | 8.3 | 8.3 |
| American Indian Female | 43 | 25.6 | 18.6 | 13.9 | 9.3 |
| Other Male | 213 | 54.0 | 32.9 | 25.3 | 22.5 |
| Other Female | 384 | 39.1 | 26.0 | 21.6 | 18.0 |

*Notes.* M1 = Milestone 1, including a first semester of general chemistry, biology, and physics, M2 = Milestone 2, including two semesters of general chemistry, biology, and physics, M3 = Milestone 3, including Milestone2 as well as a first semester of organic chemistry, M4 = Milestone 4 (full fulfillment), including Milestone 3 as well as a second semester of organic chemistry or a semester of biochemistry. American Indian = American Indian or Alaska Native, Asian = Asian, Asian American, or Pacific Islander, Black = Black or African American, Hispanic = Mexican or Mexican American, Puerto Rican, or Other Hispanic, Latino, or Latin American, Other = Other minorities.

coursework definitionally distinguishes fulfillers from non-fulfillers. They were also found to have performed better in these courses (average $d = .42$).

## Logistic regression models for predicting fulfillment progression

Logistic regression analyses were performed for the four fulfillment milestones. Predictors included gender, ethnicity, the composite SAT score, high school GPA, SES, and mean college pre-med course grades when applicable. Continuous variables—SAT, GPA, SES, and course grades—were standardized to aid the comparison between regression coefficients. Variable intercorrelations with each subset of the sample can be found in S1, S2, S3 and S4 Tables in the supplement. The progression of logistic regression models can be found in Table 3. The same models with individual prior course grades as predictors were also tested and yielded similar results.

**Table 2. Mean (SD) for continuous variables for fulfillers and non-fulfillers: 2006–2009.**

| Variables | Fulfillers | | | Non-Fulfillers | | | |
| --- | --- | --- | --- | --- | --- | --- | --- |
| | N | Mean | SD | N | Mean | SD | d |
| SAT-Verbal | 2,516 | 601.60 | 84.45 | 12,748 | 572.45 | 87.06 | 0.34 |
| SAT-Math | 2,516 | 636.87 | 81.81 | 12,748 | 591.94 | 89.32 | 0.51 |
| SAT-Writing | 2,508 | 599.53 | 86.62 | 12,727 | 571.09 | 88.67 | 0.32 |
| SAT-Composite | 2,508 | 1838.05 | 222.62 | 12,727 | 1735.54 | 236.47 | 0.44 |
| High School GPA | 2,545 | 3.96 | 0.32 | 12,786 | 3.83 | 0.39 | 0.33 |
| SES | 2,471 | 0.34 | 0.94 | 12,504 | 0.17 | 0.97 | 0.18 |
| AP grades | | | | | | | |
| AP Biol | 1,167 | 3.75 | 1.20 | 4,204 | 3.06 | 1.39 | 0.52 |
| AP Chem | 817 | 3.18 | 1.29 | 2,376 | 2.68 | 1.35 | 0.38 |
| AP Phys B | 303 | 3.14 | 1.24 | 959 | 2.75 | 1.29 | 0.31 |
| AP Phys E | 70 | 3.73 | 1.24 | 135 | 2.88 | 1.42 | 0.62 |
| AP Phys M | 146 | 3.74 | 1.11 | 346 | 3.01 | 1.33 | 0.57 |
| Number of courses taken | | | | | | | |
| Biochem N | 2,555 | 0.66 | 0.86 | 12,887 | .20 | 0.51 | 0.32 |
| Biol N | 2,555 | 5.99 | 3.00 | 12,887 | 2.56 | 2.51 | 0.65 |
| GenChem N | 2,555 | 2.81 | 1.34 | 12,887 | 1.54 | 1.28 | 0.53 |
| OrgChem N | 2,555 | 2.16 | 0.93 | 12,887 | 0.44 | 0.80 | 0.83 |
| Phys N | 2,555 | 2.35 | 0.79 | 12,887 | 0.72 | 1.24 | 0.77 |
| College grades | | | | | | | |
| Biochem GR | 1,178 | 3.15 | 0.83 | 2,020 | 2.88 | 0.94 | 0.30 |
| Biol GR | 2,547 | 3.16 | 0.68 | 9,754 | 2.71 | 0.90 | 0.52 |
| GenChem GR | 2,545 | 3.16 | 0.69 | 9,382 | 2.66 | 0.95 | 0.56 |
| OrgChem GR | 2,547 | 2.90 | 0.85 | 3,746 | 2.58 | 0.96 | 0.35 |
| Phys GR | 2,524 | 3.20 | 0.75 | 4,599 | 2.88 | 0.94 | 0.37 |

*Note*. Fulfillment = two semesters of general chemistry, biology, physics, and organic chemistry or one semester of organic chemistry with one semester of biochemistry, d = Cliff's $\delta$ for number of courses and Cohen's d for all other variables, SES = socio-economic status, Biol = Biology, Chem = Chemistry, Phys B = Physics, Phys E = Physics C Electricity & Magnetism, Phys M = Physics C Mechanics, Biochem = Biochemistry, GenChem = General Chemistry, OrgChem = Organic Chemistry.

Table 3 reports the odds ratios (OR) and their 95% confidence intervals (CI) for various milestones. To aid the interpretation of these results, we also computed predicted likelihoods based on the regression weights. To compute fulfillment likelihoods of various demographic groups, we plugged in average values of all other predictors. To compute the continuous variables' effects on fulfillment likelihoods, we plugged in the reference groups (i.e., females for gender, White for race), average values of all other predictors, and the average and 1 standard deviation (SD) above average values of the predictor of interest.

Model 1 examined whether students took first-semester general chemistry, biology, and physics courses with demographic and pre-college predictors (milestone 1). The predicted likelihood of males fulfilling milestone 1 was 8.50% higher than that of females (odds ratio (OR) = 1.44), and the likelihood of Asian students completing the milestone was 11.97% higher than that of White students (OR = 1.65), net of all other predictors. Further, controlling for all other predictors, the predicted likelihood of fulfilling milestone 1 increased by 4.82% with every 1 SD increase in SAT score (OR = 1.23), 5.63% with every 1SD increase in high school GPA (OR = 1.28), and .98% with every 1SD increase in SES (OR = 1.04).

Model 2 examined whether students who completed a first semester of prerequisites proceeded to complete a second semester of general chemistry, biology, and physics (milestone 2),

**Table 3. Adjusted odds ratios (OR) and their 95% confidence intervals (CI) for various milestones towards pre-med course fulfillment: 2006–2009.**

| Predictor | Milestone 1 | | Milestone 2 | | Milestone 3 | | Full Fulfillment (Milestone 4) | |
|---|---|---|---|---|---|---|---|---|
| | OR | 95% CI | OR | 95% CI | OR | 95% CI | OR | 95% CI |
| Intercept | 0.50** | [0.47, 0.53] | 1.52** | [1.38, 1.68] | 3.48** | [3.04, 4.00] | 3.59** | [3.07, 4.21] |
| Gender: Male | 1.44** | [1.33, 1.55] | 1.21** | [1.08, 1.36] | 1.15 | [0.98, 1.35] | 1.18 | [0.97, 1.44] |
| Race: Black | 1.05 | [0.93, 1.17] | 1.15 | [0.93, 1.43] | 0.86 | [0.64, 1.15] | 0.74 | [0.52, 1.05] |
| Race: Asian | 1.65** | [1.49, 1.82] | 1.07 | [0.94, 1.23] | 1.02 | [0.84, 1.24] | 0.93 | [0.75, 1.16] |
| Race: Hispanic | 1.04 | [0.92, 1.17] | 1.44** | [1.17, 1.80] | 0.64** | [0.50, 0.83] | 1.17 | [0.82, 1.67] |
| SATC | 1.23** | [1.19, 1.28] | 0.95 | [0.88, 1.03] | 1.33** | [1.20, 1.46] | 0.88* | [0.78, 1.00] |
| HSGPA | 1.28** | [1.22, 1.32] | 0.97 | [0.92, 1.03] | 1.17** | [1.07, 1.26] | 1.03 | [0.93, 1.14] |
| SES | 1.04* | [1.00, 1.08] | 0.95 | [0.90, 1.01] | 0.95 | [0.88, 1.03] | 1.01 | [0.92, 1.11] |
| Grades | | | 1.56** | [1.46, 1.65] | 1.07 | [0.97, 1.18] | 1.21* | [1.05, 1.39] |
| OChem | | | | | | | 0.99 | [0.86, 1.14] |

*Note*. Reference group for gender was female and for race was white. Milestone1 = whether students took a first semester of general chemistry, biology, and physics, Milestone2 = whether students who completed Milestone1 took a second semester of general chemistry, biology, and physics, Milestone3 = whether students who completed Milestone2 took a first semester of organic chemistry, Full Fulfillment (Milestone4) = whether students who completed Milestone3 took a second semester of organic chemistry or a semester of biochemistry, CI = confidence interval, Race: Black = Black or African American, Race: Asian = Asian, Asian American, or Pacific Islander, Race: Hispanic: Mexican or Mexican American, Puerto Rican, or Other Hispanic, Latino, or Latin American, SATC = standardized SAT composite, HSGPA = standardized self-reported high school GPA, SES = standardized socio-economic status, Grades = standardized average grade of all prior courses in general chemistry, biology, and physics, OChem = standardized grade in first-semester organic chemistry.

*$p < .05$.

**$p < .001$.

using not only demographic and pre-college predictors, but also the mean of first-semester general chemistry, biology, and physics grades. Among students who had fulfilled milestone 1, the predicted likelihood of males fulfilling milestone 2 was 4.48% higher than that of females (OR = 1.21), and the likelihood of Hispanic students was 8.34% higher than that of White females (OR = 1.44) when holding all other predictors constant. Further, 1SD increase in average first-semester course grades improved the likelihood of students fulfilling milestone 2 by 10.04% net of all other predictors (OR = 1.56).

Subsequently, model 3 used the same set of predictors with course grades computed as the average of grades across both semesters of general chemistry, biology, and physics to determine whether students who have fulfilled milestone 2 took any organic chemistry courses (milestone 3). Among students who had completed milestone 2, the likelihood of Hispanic students taking and passing an organic chemistry course was 8.58% lower than that of White females when holding all other predictors constant at average (OR = .64). The likelihood of fulfilling milestone 3 also increased by 4.51% with every 1SD increase in SAT score (OR = 1.33) and by 2.54% with every 1SD increase in high school GPA (OR = 1.17), net of all other variables.

Finally, Model 4 used the same previous set of predictors along with the first organic chemistry grade to predict whether students took a second organic chemistry course or a biochemistry course, thereby fulfilling all science coursework prerequisites for medical school, conditional on having completed all required courses thus far (milestone 4). There were no statistically significant difference in fulfillment likelihood between gender and racial majority and minority groups. Controlling for all other predictors, 1SD increase SAT scores decreased predicted likelihood of prerequisite completion by 2.24% (OR = .88), and 1SD increase in average coursework grades increased likelihood by 3.09% (OR = 1.21).

To explore the potential demographic variables' interactions in predicting coursework fulfillment likelihoods, the same four models were also tested including the interaction terms between gender and race dummy variables. The only statistically significant and meaningful interaction was between being male and being Asian for fulfilling coursework milestone 2 (OR = .71, $p < .001$) and milestone 3 (OR = .69, $p < .001$). In other words, holding all other predictors constant at their average values, while the predicted likelihood of White males fulfilling milestone 2 exceeds that of White females by 8.67%, the difference in likelihood between Asian males and females is only 2.14%. Similarly, the difference between White males' and females' predicted likelihood for completing milestone 3 is 7.81%, while that between Asian males and females is 1.53%.

Additional logistic regression models with various interactions terms between demographic dummy variable (i.e., gender and race) and the continuous predictors (i.e., SAT, high school GPA, SES, and grades) were also tested individually. For predicting milestones 1, 2, and 3, there was a significant interaction between gender and SAT score such that the difference in fulfillment likelihood between males and females (in favor of males) was reduced as SAT score increased (OR = .88, .84, and .88, respectively). A similar effect was found in the White–Asian comparison for milestones 1 and 2, such that the higher fulfillment likelihood of Asian students than White students was reduced with higher SAT scores (OR = .83 and .84, respectively). In addition, for predicting fulfillment of milestone 3, the higher likelihood of Asian than White students was reduced as average college course grade increased (OR = .88). Lastly, the higher likelihood of Asian students fulfilling milestone 4 than White students was enhanced with higher high school GPA (OR = 1.31).

## Discussion

The current study examines the pre-medical coursework fulfillment patterns of the group of students who indicated intentions of studying pre-medicine prior to entering college. Only 16.5% of the students graduated with the coursework required by most medical schools. Attrition is highest at early stages and levels off as students commit to the medical education track by taking more of the required courses. Previous studies that found that former pre-med students often mentioned a "distaste for the large pre-med classes" and the highly competitive environment [3], and a change in interest as a result of exposure to other subjects [3,12,19]. Thus, while attrition rates in the later college years are comparatively lower and may be attributed to challenging coursework, the initial high attrition may reflect students adjusting their expectations about medicine while discovering interest in non-medical disciplines. This is also consistent with earlier findings that students change their majors often due to interest in and positive perceptions of new major more than negative factors about the old major [28]. Given the low acceptance rates into medical schools [6] and the attrition rates in medical schools [4,29], this early change in education track may actually prevent additional personal resources from being wasted in the process of applying to medical programs or institutional resources from being wasted when students drop out of medical programs.

Although a much higher percentage of intended pre-med students completed the full set of medical school prerequisite courses and ended up eligible for medical schools (16.5%) than students with no initial intent (3.9%), the absolute number of the latter group (2,633) was comparable with that of the former group (2,555). The 2019 MSQ by the AAMC reported that of the medical school matriculants, 55.3% had decided that they wanted to study medicine prior to entering college and 34.8% did so during college [6]. When interpreted in light of the present findings, it is evident that a higher percentage of the students who completed medical school prerequisite coursework with initial intent is accepted than without initial intent.

## Predictors of pre-med persistence

Similar to previous findings on the association between gender and attrition [3,10] and consistent with the normative alternatives approach to explaining the persistence gap [9], being male was linked with a significantly higher likelihood of persisting in a pre-medical education at nearly all stages throughout college. In other words, women's lower likelihood to persist in medicine may be construed as a higher likelihood to accept alternative career choices.

With regards to ethnic and racial identities, while previous investigations suggest that ethnic and racial minority students are less likely to persist [3,8], results of the current study were less consistent. For Asian students, the odds of fulfilling the first semester of coursework were more likely than those of White students, with persistence likelihood decreasing slightly throughout the later milestones but never significantly lower than those of White students. The odds of Hispanic students fulfilling the first year of coursework were higher than those of White students, but the odds of their completing organic chemistry were lower. Lastly, African American students did not differ from White students in their likelihood of persistence at any point after controlling for socio-economic status, SAT, and grades in high school and college.

Further, students who fulfilled all required coursework reported higher SES. Although SES predicted completion of the first semester of required coursework, it did not predict persistence at any of the later milestones. Thus, the advantage of coming from a family with higher SES found by [3] seems to wear off early on during college. However, its link with persistence in a medical education is likely to strengthen when students decide whether or not to attend medical school, as a $200,000 to $300,000 cost for medical school is no small expense [30].

Consistent with the reputation of a degree in pre-medical studies for being cognitively intensive and challenging, coursework fulfillers entered college with higher scores on all components of the SAT as well as higher high school GPA. Continuing with this advantage, college GPA's of students who eventually fulfilled all required coursework were higher than the GPA's of those who did not for all four years. However, the differences declined over time, with the GPA difference in the fourth year of college being half as large as the GPA difference in the first year of college. This may be an indirect reflection of the high levels of difficulty in pre-medical courses compared with other courses. In terms of college performance, coursework fulfillers both by definition completed a larger number of courses in relevant science subjects and obtained higher grades in them than non-fulfillers.

When examining the predictive validities of academic preparedness measured by variables prior to college (SAT and high school GPA) and college performance (course grades), an interesting pattern is observed. For predicting completion of the first semester of coursework in the absence of any college grades, higher academic preparedness was associated with a greater likelihood of completion. However, the predictive validities of such more distal, pre-college factors were overtaken by that of the more proximal college grades. The exception is whether students complete the first organic chemistry, for which academic preparedness rather than college grades was predictive. This may be due to the notoriously difficult organic chemistry being overwhelmingly identified as culprit in the leaky pipeline [3,11]. Students might be extra cautious in deciding whether they would be able to succeed in the course and resort to information about their academic effectiveness over a longer period of their lives from the past rather than grades from the recent one or two years to make the decision. In a recent study examining persistence in undergraduate STEM courses, it was found that grades in the first general chemistry course was related with subsequent persistence in STEM more strongly among underrepresented individuals than among well-represented students [20]. Our examination of such interactions between demographic group and other predictors (e.g., academic preparedness before college, grades in college) yielded mixed results.

## Limitations and future directions

There are several limitations in the current study. First, because we lacked information about whether students actively pursued pre-medicine throughout college, we focused on completion of coursework required for medical programs. This operationalization is indirect and imperfect. It is possible that students of other science majors completed a similar set of coursework, cases that were considered noise in the current study. On the other hand, it is also possible that some students do not or partially complete prerequisite coursework during their undergraduate years, but later complete all required courses in a postbaccalaureate premedical program. A large number of such programs already existed during the time that data used in the current study was collected [31]. Thus, by focusing on students' medical school eligibility in terms of their coursework at their undergraduate institution, the experiences of those that only become eligible later on were not captured.

Second, as we were not provided information about whether students who obtained satisfactory scores on AP exams actually used their AP credits toward their college degree, we assumed that any AP exam score of 3 or higher counted as a fulfillment course. There is the possibility that some students may forfeit their AP credits or take the equivalent college course.

Third, since the collection of data used in the present study, a new version of the MCAT that includes a larger social and behavioral science component has been implemented [32]. It is reasonable to assume that requirements by medical schools were also revised to contain psychology and sociology coursework. It will be valuable to examine the effects of these changes. Furthermore, major shifts have taken place with regards to the philosophy underlying the evaluation of medical school applicants. In the past decade, AAMC has endorsed the value of "holistic review," which considers "applicants' experiences, attributes, and academic metrics as well as the value an applicant would contribute to learning, practice, and teaching" [33]. As such, academic record and performance is considered alongside many other factors for admission decisions, such as "distance traveled" or cumulative life experiences, and other contextual information for the applicants' accomplishment [34–36]. In response to the call for holistic evaluation, a number of medical schools have revised their admissions statements and requirements. For example, the Perelman School of Medicine at the University of Pennsylvania define competencies "not based on specific courses, but rather on the cumulative achievement of knowledge and skills needed to become a physician" [37]. The Boston University School of Medicine emphasize "experiential and personal qualities" in addition to academic rigor in their selection process [38]. There are even institutions like Stanford University School of Medicine that explicitly removed any specific prerequisite requirements, and only provide course recommendations instead [39]. Since the data used in the current study was collected prior to these changes, it will be important for future research to differently operationalize and examine pre-med intention and persistence.

Finally, as our study relied on archival data, we were limited in the extent to which we could explore underlying mechanisms that explain attrition in pre-medical studies. While the regression analyses revealed certain demographic and academic preparedness factors as predictors, no information about the specific reasons behind students' dropping out was available. However, our study provides a valuable quantitative complement to the prior qualitative body of research that described such reasons.

## Conclusion

The present study quantitatively describes the process of attrition as reflected in coursework throughout pre-medical studies in postsecondary institutions. A number of pre-college preparedness factors including socio-economic status, SAT score, and high school GPA as well as

grades during college were found to predict continued eligibility for medical studies at at various milestones of relevant coursework.

## Supporting information

**S1 Table. Variable intercorrelations using sub-sample included in the logistic regression model for Milestone 1.**
(DOCX)

**S2 Table. Variable intercorrelations using sub-sample included in the logistic regression model for Milestone 2.**
(DOCX)

**S3 Table. Variable intercorrelations using sub-sample included in the logistic regression model for Milestone 3.**
(DOCX)

**S4 Table. Variable intercorrelations using sub-sample included in the logistic regression model for Milestone 4.**
(DOCX)

## Author Contributions

**Conceptualization:** Nathan R. Kuncel, Paul R. Sackett.

**Formal analysis:** Charlene Zhang.

**Funding acquisition:** Nathan R. Kuncel, Paul R. Sackett.

**Investigation:** Charlene Zhang, Nathan R. Kuncel.

**Methodology:** Charlene Zhang, Nathan R. Kuncel, Paul R. Sackett.

**Project administration:** Charlene Zhang.

**Supervision:** Nathan R. Kuncel, Paul R. Sackett.

**Validation:** Charlene Zhang.

**Visualization:** Charlene Zhang.

**Writing – original draft:** Charlene Zhang.

**Writing – review & editing:** Nathan R. Kuncel, Paul R. Sackett.

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
