## [Decision Letter · Decision Letter 0]

30 Sep 2020

PONE-D-20-15864

The process of attrition in pre-medical studies: A large-scale analysis across 102 schools

PLOS ONE

Dear Dr. Zhang,

Thank you for submitting your manuscript to PLOS ONE. After careful consideration, we feel that it has merit but does not fully meet PLOS ONE’s publication criteria as it currently stands. Therefore, we invite you to submit a revised version of the manuscript that addresses the points raised during the review process.

Your paper received comments from four reviewers. They all found the theme of the paper important. However, they shared several concerns with the paper. I am particularly concerned the missing issues in the paper as these issues if addressed could improved the quality of the paper's message.

We look forward to receiving your revised manuscript.

Kind regards,

Luisa N. Borrell, DDS, PhD

Academic Editor

PLOS ONE

Journal Requirements:

Reviewers' comments:

Reviewer's Responses to Questions

**Comments to the Author**

1. Is the manuscript technically sound, and do the data support the conclusions?

Reviewer #1: Yes

Reviewer #2: Yes

Reviewer #3: Yes

Reviewer #4: Partly

2. Has the statistical analysis been performed appropriately and rigorously? 

Reviewer #1: Yes

Reviewer #2: Yes

Reviewer #3: Yes

Reviewer #4: Yes

3. Have the authors made all data underlying the findings in their manuscript fully available?

Reviewer #1: Yes

Reviewer #2: No

Reviewer #3: Yes

Reviewer #4: No

4. Is the manuscript presented in an intelligible fashion and written in standard English?

Reviewer #1: Yes

Reviewer #2: Yes

Reviewer #3: Yes

Reviewer #4: Yes

5. Review Comments to the Author

Reviewer #1: This manuscript provides a valuable analysis of the academic trajectories of students who enter college with an interest in pursuing premedical studies, and subsequently to attend medical school. Using a large data set provided by the College Board, the authors were able to identify students who had expressed the interest in premedical studies based on students’ response to a question administered at the time students took the SAT exam. As the authors indicate, more than 15,000 students indicated the intent to pursue premedical studies once they entered college.

The metric the authors utilize to gauge students’ continued level of interest in being premed is the extent to which students complete the science courses that have traditionally been required by medical schools for entry. They analyze the successful completion of these courses in a stepwise manner, as described in the Methods section. Of the students included in the data set, 16.5% completed the full sequence of courses identified by the authors by the time they had completed their undergraduate education. The authors then did an analysis of the academic and demographic factors associated with full completion of the course sequence. They identified gender, race/ethnicity, and prior academic record as among the factors associated with completion of the full course sequence.

This manuscript provides valuable information, potentially useful to those who advise undergraduates in career options. The data set the authors use is unique in its size and the variables it contains. As such, the manuscript adds value to our knowledge of the factors associated with persistence in interest in premedical studies, which is an important factor in the eventual diversity of the physician workforce.

The manuscript reports data on students who entered college between 2006 and 2009. Most of these students would have completed their undergraduate education by 2013-2014. As such, the analysis has missed some fundamental changes that have taken place in the premedical experience. As a result, some of their assumptions and findings may need re-examination by the authors. As one example, the authors state in the Abstract, “Attrition rates are highest initially but drop as students take more relevant courses.” The issue is not the relevance of the courses. The issue is how advanced the courses are. The AAMC has done research indicating that inorganic chemistry courses have more relevance to the practice of medicine than do organic chemistry courses. Organic chemistry courses are more advanced than inorganic chemistry courses, but they are not more “relevant”.

A second thing the authors miss in their discussion is the fundamental shift that has been taking place in the medical school admission process, with the shift to holistic review. In April 2013, Witzburg and Sondheimer published an article in the New England Journal of Medicine titled, “Holistic Review — Shaping the Medical Profession One Applicant at a Time.” Under holistic review, as described by the AAMC, a student’s grades in the premedical science courses, while still relevant, are only one of a series of factors used in selecting candidates for medical school. Consistent with the holistic review model, a growing number of medical schools no longer have a specific list of prerequisite courses required for admission. For example, the Perelman School of Medicine at the University of Pennsylvania reports on its Admissions website, “Admissions competencies are not based on specific courses, but rather on the cumulative achievement of knowledge and skills needed to become a physician.” (https://www.med.upenn.edu/admissions/admissions.html)

Similarly, the Admissions webpage of the Stanford University School of Medicine states explicitly, “Stanford Medicine does not have specific course requirements, but recommends appropriate preparation for the study of medicine.”

(http://med.stanford.edu/md-admissions/how-to-apply/academic-requirements.html)

Those medical schools that have dropped their explicit course requirements are instead relying on the newly formatted MCAT to provide a metric of scientific and behavioral competencies. Accordingly, in order for this manuscript to sustain its relevance to the current premedical experience, the authors should include a full discussion of the changes that have taken place since their study cohort graduated from college.

There is another important factor the authors have left out of their analysis and discussion. While most students who enter college with an intent to pursue premedical studies will complete the sequence of science courses at their undergraduate institution, a substantial number will elect to take either no science courses or some science courses at their undergraduate institution, while completing the full list of traditional science courses as part of a postbaccalaureate premedical program. The AAMC currently identifies 267 programs nationally (https://apps.aamc.org/postbac/#/index) that offer students the opportunity to complete their premedical course sequence after they have graduated from their undergraduate institution. Most of these programs have been active for more than a decade, and thus were available to the students included in the authors’ data set. The authors are unable to identify which of the students elected to take only some courses as an undergraduate, while completing the course sequence in a past-bac program.

I should point out that, due to poor quality graphics of Figure 1 in the manuscript, I was unable to decipher the figure. The text and images were so faint that I could not read them. I would ask the authors to revise the figure with improved graphics.

As one final issue, I suggest the authors include in their discussion the results of a paper that was published this month that reported on the association between grades in undergraduate chemistry classes and persistence in STEM majors:

R. B. Harris et al. Reducing achievement gaps in undergraduate general chemistry could lift underrepresented students into a “hyperpersistent zone”. Science Advances. 10 Jun 2020:

Vol. 6, no. 24, eaaz5687 DOI: 10.1126/sciadv.aaz5687

Reviewer #2: 0. This paper is important for empirically investigating the 'pre-med pipeline,' where many students enter into pre-med (an unspecified 'major' at many schools, including mine) -- and many drop out. Who are these students, and how does dropout unfold over time? That is the focus of the work, which should be of great interest to a wide range of stakeholders: e.g., universities, students and parents, STEM researchers, and medical educators and institutions. Authors have large-scale transcript data across 102 institutions; they have students' major intentions prior to college enrollment -- an impressive data set to answer these questions.

1. A passing grade (.7 GPA or higher) was used as a screen for eligibility to continue taking pre-med courses, but authors might clarify how this was used (being eligible is necessary but not sufficient for taking the next course). Related to this (but entirely optional), authors might indicate % failure (and didn't retake) of initial courses by demographics as one barrier to entry; and/or investigating course grades more generally might predict staying in pre-med.

2. p. 13 - One can easily appreciate how SAT, HSGPA, and CGPA would be higher for course-fulfillers, to the extent non-fulfilling is driven by obtaining low grades (or failures) in pre-med courses; this would suggest examining or at least suggesting course grades as an explanatory mechanism (see previous point). Regarding SES differences, could they be higher if college dropouts were included (those who couldn't afford to continue at all, let alone in pre-med)? Dropout might be a significant problem for pre-med and STEM majors, and for URM students, but the current sample does not include this group (maybe include this point somewhere in the discussion section, perhaps with some data on % dropout in the College Board student data).

3. Table 3 and related text - (a) interpretations of odds ratios need to be refined (they are with respect to the reference group of white females, not overall; they also need to take other predictors into account); (b) odds ratios can be usefully compared against base rate %s from Table 2 and the raw data (e.g., plug in values into the equation to get predicted probabilities, to compare against base rate%s); (c) the two previous points get at whether the pseudo-R2 values are practically useful; but in addition to this, I'd recommend a formal statistical comparison between adjacent models (e.g., compare AIC values).

4. A bit surprisingly, no institutional differences were provided -- even just adding something such as the distribution of %fulfill by institution would be interesting to readers (and this could be relatively anonymous, given 102 institutions).

Side notes:

5. Table 1 - proportion row/column information was useful, but the labeling was a little confusing

6. Table 2 - (a) the # of courses help define the fulfill/non-fulfill categories so the d-values may not be surprising (if you include them, also check out the distributions of N, as maybe there's positive skew and you consider a more robust d-value involving medians); (c) format the rows a little more descriptively (headers? better labels?)

Reviewer #3: This study examines a key outcome in the medical education track, and brings to light quantitative research findings that should provide a valuable contribution to the literature for policy-makers and admissions officers alike. The submission is clearly structured, and the analyses are appropriate to the research question, which complements existing work with a more quantitative and large scale data-based approach. I only have a handful of comments.

1) The limitations mentioned in the study are appropriate to mention, but I would have liked to also see further discussion on the relevance of data that is between 11 and 14 years old to the world of medical education today. How stable are the trends that are noted in comparison to the trends of today? Beyond noting the structural changes to MCAT, are there other factors that would lead to caution in drawing strong conclusions from this data, such as the final sentence of the conclusion?

2) When comparing fulfillers and non-fulfillers of medical school prerequisites, the paper highlights and contextualizes the larger proportion of male fulfillers, but is there a clarifying sentence that can also be added regarding meaningful differences with regards to race/ethnicity?

3) Examining subgroup differences in attrition is a key valuable component of the study, and to that end the paper would be improved by some examination or at the very least mention of interactive effects between race/ethnicity and gender and attrition. For example, the report Altering the Course: Black Males in Medicine highlights the percentage of black male medical school applicants as the lowest among any subgroup. Would conclusions such as ““Lastly, African American students did not differ from White students in their likelihood of persistence at any point.” (p 19) differ when differentiating between black men and black women?

Reviewer #4: This study investigates the completion rates of students completing the sequence of undergraduate natural science coursework prerequisite for entry to medical school in the U.S. using a large-scale, multi-institution dataset. It examines the associations of multiple sociodemographic, pre-college achievement, career aspiration, and college-level achievement measures in relation to students’ completion of four milestone events defined by natural science coursework from the first semester through advanced organic or biochemistry courses.

The study showed that, of those who reported intending to pursue a medical degree when they took the SAT in high school, only 39% completed the first in a sequence of courses that together provide the foundation needed to be ready for medical school. Completion rates were progressively higher for the (likely) increasingly select group of students achieving each of the three subsequent milestones of coursework. In the end, the overall completion rate of all four milestones was 16.5%. Observed proportions completing the full sequence were higher for males and Asian students, and lower for females and students from races/ethnicities underrepresented in medicine (i.e., American Indian, Black, and Hispanic).

High school GPA, SAT scores, SES, Gender (Male), and Race (Asian) were associated with greater likelihoods of completing the first set of milestone courses. Gender (Male), Race (Hispanic), and college grades in the first milestone courses were associated with greater likelihoods of completing the second milestone courses, which included the second semester courses of the subjects comprising the first milestone. SAT scores and high school GPA were associated with greater likelihoods of completing the third milestone courses, and race (Hispanic) with a lower likelihood. Finally, gender (Male) and intention to pursue medical school were associated with greater likelihoods of completing the fourth milestone courses, while SAT scores were associated with a lower likelihood of completing the fourth milestone courses.

Although this study has the potential to enhance our understanding of the points at which those interested in the medical profession drop out, the paper fails to make a strong case for it.

1. The authors should consider rewriting the introduction and discussion with a clearer focus and relevant citations, considering the following as suggestions.

• Integrating studies about the factors that contribute to a lack of diversity in STEM or medicine that occur during or before college with the present study, including those referenced in the current about “the leaky pipeline.” References 4, and 11-19 seem germane to the study’s focus.

• Addressing, alongside things like gender norms that might affect persistence, comparable treatment of the lower rates of completion for those from racial/ethnic backgrounds underrepresented in medicine who, more often have lower-quality middle and high school education. Addressing the role that high school preparation might play in students first-semester performance in natural science coursework and beyond would strengthen the paper.

• Removing or clarifying the relevance of literature on burnout/risk of attrition in medical school (references 3, and 5-9), and references 3, 10, and 20, which about attrition in the medical school in the U.K., which differs substantially from medical school in the U.S. The authors also should consider eliminating references about attrition in U.S. medical school (given that 95% of medical students graduate within five years) or make a stronger case for their relevance.

2. With a clearer focus, the authors might present their results differently to highlight important findings. For example, Table 1 shows that Black and Hispanic students completed all four milestones at lower rates than White or Asian students. It might be important, given the research questions, to know at which milestone(s) they did not progress, in addition to showing their prevalence at the two end points. Similarly, it was confusing on page 19 to read that African American students did not differ from White students in their likelihood of persistence at any point, even though only 9% of Black students completed all four milestones, compared to 16% of White students. The authors should address how the observed and predicted rates of completion lead to different interpretations.

3. Similar suggestions would improve the discussion of the paper’s findings.

4. Finally, the conclusions are unfounded and should be rewritten.

5. Other minor suggestions include:

• addressing how the increasingly select sample of students might affect the results of the logistic regression analyses. For example, SES, SAT scores, and high school GPA may be restricted as the sample reduces from more than 13,000 to about 2,500 students.

• Replacing outdated references with more recent research (e.g., 18) and confirming the appropriateness of journals cited (e.g., 9, which is missing the Journal Name “Journal of Unschooling and Alternative Learning”).

• Clarifying the relevance or eliminating the text describing challenges identifying premed majors. The study documented the process for identifying courses prerequisite for medical school (and that are well described on each medical school’s website).

6. PLOS authors have the option to publish the peer review history of their article (what does this mean?). If published, this will include your full peer review and any attached files.

Reviewer #1: No

Reviewer #2: No

Reviewer #3: No

Reviewer #4: No

---

## [Author Response · Author response to Decision Letter 0]

23 Oct 2020

Your paper received comments from four reviewers. They all found the theme of the paper important. However, they shared several concerns with the paper. I am particularly concerned the missing issues in the paper as these issues if addressed could improved the quality of the paper's message.

Dear Dr. Borrell,

 Thank you for initiating reviews on our paper “The process of attrition in pre-medical studies: A large-scale analysis across 102 school” and giving us the opportunity to revise and resubmit. We appreciated the reviews and constructive comments provided by the reviewers and yourself. 

 It was encouraging to see that reviewers shared our view in the importance of this work. We had a unique opportunity to quantitatively examined the process through which undergraduate students fulfill prerequisite coursework for medical schools using a large multi-cohort, multi-institution sample. Given the popularity of the informal pre-med major and little prior work focusing on attrition in the early pre-medical study stage, we felt that this work is valuable in adding to the current higher education literature.

 In our response to reviewer comments, we have made effort to address every reviewer comment point-by-point and to incorporate the feedback into our revised manuscript. Notably, we substantially restructured our Table 1 to describe the key fulfillment rates in a way that is clearer and also include rates at the intersection of gender and race and at various fulfillment milestones. We also tested additional models that included interaction terms between the various predictors. The results were not tabled given the large number of complex models, but we described key findings. Lastly, we added discussions of the recent shift to holistic evaluation of applicants in medical education and its implications for the ongoing relevance of our results. 

 A minor detail to note is that while “degree goal” was a predictor included in our original submission because it was a variable that existed in the archival data given to us, and has often been found to be predictive of academic performance in past research. However, upon reexamination, we deemed it irrelevant to the current work. We define our total sample as individuals who have intention of becoming physicians, therefore their “degree goal,” or the highest degree they hope to pursue, had very little variance. We excluded it in the revised manuscript, but none of the reported findings changed. 

 We believe that these edits, along with other minor revisions, have significantly improved our manuscript. Thank you again for your consideration. 

Reviewer #1: This manuscript provides a valuable analysis of the academic trajectories of students who enter college with an interest in pursuing premedical studies, and subsequently to attend medical school. Using a large data set provided by the College Board, the authors were able to identify students who had expressed the interest in premedical studies based on students’ response to a question administered at the time students took the SAT exam. As the authors indicate, more than 15,000 students indicated the intent to pursue premedical studies once they entered college.

The metric the authors utilize to gauge students’ continued level of interest in being premed is the extent to which students complete the science courses that have traditionally been required by medical schools for entry. They analyze the successful completion of these courses in a stepwise manner, as described in the Methods section. Of the students included in the data set, 16.5% completed the full sequence of courses identified by the authors by the time they had completed their undergraduate education. The authors then did an analysis of the academic and demographic factors associated with full completion of the course sequence. They identified gender, race/ethnicity, and prior academic record as among the factors associated with completion of the full course sequence.

This manuscript provides valuable information, potentially useful to those who advise undergraduates in career options. The data set the authors use is unique in its size and the variables it contains. As such, the manuscript adds value to our knowledge of the factors associated with persistence in interest in premedical studies, which is an important factor in the eventual diversity of the physician workforce.

We appreciate the positive feedback on the value of this work. We agree with the reviewer that this dataset provides a unique opportunity to examine the academic trajectories of pre-medical students. 

The manuscript reports data on students who entered college between 2006 and 2009. Most of these students would have completed their undergraduate education by 2013-2014. As such, the analysis has missed some fundamental changes that have taken place in the premedical experience. As a result, some of their assumptions and findings may need re-examination by the authors. As one example, the authors state in the Abstract, “Attrition rates are highest initially but drop as students take more relevant courses.” The issue is not the relevance of the courses. The issue is how advanced the courses are. The AAMC has done research indicating that inorganic chemistry courses have more relevance to the practice of medicine than do organic chemistry courses. Organic chemistry courses are more advanced than inorganic chemistry courses, but they are not more “relevant”.

We have changed “relevant” to “advanced” in the abstract as we agree that later, more advanced courses are not necessarily more relevant that earlier courses. We use the description “relevant” at several other places throughout the manuscript to refer to all courses used to operationalize medical school eligibility that are common prerequisites. 

A second thing the authors miss in their discussion is the fundamental shift that has been taking place in the medical school admission process, with the shift to holistic review. In April 2013, Witzburg and Sondheimer published an article in the New England Journal of Medicine titled, “Holistic Review — Shaping the Medical Profession One Applicant at a Time.” Under holistic review, as described by the AAMC, a student’s grades in the premedical science courses, while still relevant, are only one of a series of factors used in selecting candidates for medical school. Consistent with the holistic review model, a growing number of medical schools no longer have a specific list of prerequisite courses required for admission. For example, the Perelman School of Medicine at the University of Pennsylvania reports on its Admissions website, “Admissions competencies are not based on specific courses, but rather on the cumulative achievement of knowledge and skills needed to become a physician.” (https://www.med.upenn.edu/admissions/admissions.html)

Similarly, the Admissions webpage of the Stanford University School of Medicine states explicitly, “Stanford Medicine does not have specific course requirements, but recommends appropriate preparation for the study of medicine.”

(http://med.stanford.edu/md-admissions/how-to-apply/academic-requirements.html)

Those medical schools that have dropped their explicit course requirements are instead relying on the newly formatted MCAT to provide a metric of scientific and behavioral competencies. Accordingly, in order for this manuscript to sustain its relevance to the current premedical experience, the authors should include a full discussion of the changes that have taken place since their study cohort graduated from college.

We thank the reviewer for pointing out the substantial shift in the evaluation process of medical school applicants. Given when our data was collected and how relatively uniform the prerequisite requirements were across medical schools during that time period, we were able to operationalize medical school eligibility using coursework. As the emphasis on prerequisites relaxes, this approach will become no longer reliable. Per the reviewer’s suggestion, we add some language in the limitations and future directions section to highlight this issue. 

There is another important factor the authors have left out of their analysis and discussion. While most students who enter college with an intent to pursue premedical studies will complete the sequence of science courses at their undergraduate institution, a substantial number will elect to take either no science courses or some science courses at their undergraduate institution, while completing the full list of traditional science courses as part of a postbaccalaureate premedical program. The AAMC currently identifies 267 programs nationally (https://apps.aamc.org/postbac/#/index) that offer students the opportunity to complete their premedical course sequence after they have graduated from their undergraduate institution. Most of these programs have been active for more than a decade, and thus were available to the students included in the authors’ data set. The authors are unable to identify which of the students elected to take only some courses as an undergraduate, while completing the course sequence in a past-bac program.

It is certainly true that our approach of identifying students with intention in studying medicine and operationalizing their continued persistence is not perfect. As the first of the limitations we pointed out, by focusing on coursework, we inevitably include those that have the required coursework for most medical schools but nevertheless lost their intention to study medicine along the way. We thank the reviewer for pointing out another group considered as noise, namely those that do not complete the required coursework during their undergraduate years, but eventually do in a postbaccalaureate program. We added language to reflect this limitation. 

I should point out that, due to poor quality graphics of Figure 1 in the manuscript, I was unable to decipher the figure. The text and images were so faint that I could not read them. I would ask the authors to revise the figure with improved graphics.

The figure has been regenerated to be higher-quality. 

As one final issue, I suggest the authors include in their discussion the results of a paper that was published this month that reported on the association between grades in undergraduate chemistry classes and persistence in STEM majors:

R. B. Harris et al. Reducing achievement gaps in undergraduate general chemistry could lift underrepresented students into a “hyperpersistent zone”. Science Advances. 10 Jun 2020:

Vol. 6, no. 24, eaaz5687 DOI: 10.1126/sciadv.aaz5687

We have added reference of the study. We found the key results of interaction between race and grades for predicting subsequent persistence intriguing. Therefore, we also tested models additional to the ones reported in Table 3 that included various interactions between demographic dummies and continuous predictors. Because this yielded 32 additional models that contained largely redundant and statistically insignificant results, they are not tabled and presented. However, we added a paragraph at the end of the Results section describing the significant interactions. 

Reviewer #2: 0. This paper is important for empirically investigating the 'pre-med pipeline,' where many students enter into pre-med (an unspecified 'major' at many schools, including mine) -- and many drop out. Who are these students, and how does dropout unfold over time? That is the focus of the work, which should be of great interest to a wide range of stakeholders: e.g., universities, students and parents, STEM researchers, and medical educators and institutions. Authors have large-scale transcript data across 102 institutions; they have students' major intentions prior to college enrollment -- an impressive data set to answer these questions.

1. A passing grade (.7 GPA or higher) was used as a screen for eligibility to continue taking pre-med courses, but authors might clarify how this was used (being eligible is necessary but not sufficient for taking the next course). Related to this (but entirely optional), authors might indicate % failure (and didn't retake) of initial courses by demographics as one barrier to entry; and/or investigating course grades more generally might predict staying in pre-med.

As we operationalized medical school eligibility as the fulfillment of various relevant courses, the passing grade was merely used as a minimum standard for which a course can be counted toward achievement. For example, the first milestone requires the completion of one semester of general chemistry, biology, and physics with a passing grade. We revised the language of the four milestones to make this clearer. 

2. p. 13 - One can easily appreciate how SAT, HSGPA, and CGPA would be higher for course-fulfillers, to the extent non-fulfilling is driven by obtaining low grades (or failures) in pre-med courses; this would suggest examining or at least suggesting course grades as an explanatory mechanism (see previous point). Regarding SES differences, could they be higher if college dropouts were included (those who couldn't afford to continue at all, let alone in pre-med)? Dropout might be a significant problem for pre-med and STEM majors, and for URM students, but the current sample does not include this group (maybe include this point somewhere in the discussion section, perhaps with some data on % dropout in the College Board student data).

We agree with the reviewer about the merit of including course grades as a predictor. Our current logistic regression models include average prior relevant course grade as a predictor when possible. We also tested but did not report results of models that included individual course grades separately, as the models currently reported are more parsimonious and yielded similar results. 

Unfortunately, the data we were given do not let us identify students that drop out. By examining coursework data, we are able to identify individuals that have coursework up to some point during their undergraduate years, but do not complete all coursework. However, we have no means of distinguishing between those that truly dropped out and those that transferred to a different institution. 

3. Table 3 and related text - (a) interpretations of odds ratios need to be refined (they are with respect to the reference group of white females, not overall; they also need to take other predictors into account); (b) odds ratios can be usefully compared against base rate %s from Table 2 and the raw data (e.g., plug in values into the equation to get predicted probabilities, to compare against base rate%s); (c) the two previous points get at whether the pseudo-R2 values are practically useful; but in addition to this, I'd recommend a formal statistical comparison between adjacent models (e.g., compare AIC values).

We thank the reviewer for urging us to report regression findings with more accurate language. We have revised the results section to specify the reference group of the dummy variables as well as the interpretation of coefficients when controlling for all other predictors. 

We also agree that odds ratios can be unintuitive and difficult to interpret, and describing the effects in terms of predicted probabilities can facilitate interpretation. Therefore, we have revised the results section by translating some of the key coefficients and odds ratios into differences in likelihood. 

We have also added AIC values of the regression models that we tested in Table 3. 

4. A bit surprisingly, no institutional differences were provided -- even just adding something such as the distribution of %fulfill by institution would be interesting to readers (and this could be relatively anonymous, given 102 institutions).

We agree with the reviewer that some attention should be paid to institutional differences. The sample sizes per institution varied substantially in the current data, ranging from 2 to 1,475. Therefore, proportions of students that fulfilled coursework milestones in institutions for which we only had sample sizes in the single digits could lead to misleading conclusions. However, we provide an added paragraph near the beginning of the results section with the mean and standard deviation of fulfillment proportion across institutions. We hope this sheds some light on the fulfillment distributions. 

Side notes:

5. Table 1 - proportion row/column information was useful, but the labeling was a little confusing

We agree that the original Table 1 could benefit from clearer labels. In addition, we received suggestions from other reviewers to expand this table to also look at the fulfillment breakdown at the intersection of gender and race, as well as to look at fulfillment proportions at the intermediate milestones. In an effort to take into consideration all of these suggestions, we completely restructured our Table 1 to provide the sample size and percentage of individuals that fulfilled each of the four coursework milestones for the entire sample, by gender, by race, and by gender and race. By doing so, we focus the purpose of the table on displaying the proportion who remain med-school-eligible at each stage, and how the proportions compare across demographic groups. We think that this significantly improves the interpretability and readability of the table. 

6. Table 2 - (a) the # of courses help define the fulfill/non-fulfill categories so the d-values may not be surprising (if you include them, also check out the distributions of N, as maybe there's positive skew and you consider a more robust d-value involving medians); (c) format the rows a little more descriptively (headers? better labels?)

The reviewer is correct in pointing out that the number of courses taken are quite different between fulfillers and non-fulfillers by definition and they are skewed. We have revised the table to show Cliff’s delta rather than Cohen’s d for those variables. Row names have been revised to be more descriptive and sub-headings have been added better describe the variables. 

Reviewer #3: This study examines a key outcome in the medical education track, and brings to light quantitative research findings that should provide a valuable contribution to the literature for policy-makers and admissions officers alike. The submission is clearly structured, and the analyses are appropriate to the research question, which complements existing work with a more quantitative and large scale data-based approach. I only have a handful of comments.

1) The limitations mentioned in the study are appropriate to mention, but I would have liked to also see further discussion on the relevance of data that is between 11 and 14 years old to the world of medical education today. How stable are the trends that are noted in comparison to the trends of today? Beyond noting the structural changes to MCAT, are there other factors that would lead to caution in drawing strong conclusions from this data, such as the final sentence of the conclusion?

We thank the reviewer for the thoughtful comment. There certainly are additional factors that might influence the generalizability and relevance of the findings. A major one is the endorsement of “holistic review” by medical applicant evaluation and the shift from primary emphasis on academic performance to a consideration of all aspects of applicants’ experiences and potential. We added some language in the limitations section explaining this shift in the past decade and its implications. 

2) When comparing fulfillers and non-fulfillers of medical school prerequisites, the paper highlights and contextualizes the larger proportion of male fulfillers, but is there a clarifying sentence that can also be added regarding meaningful differences with regards to race/ethnicity?

We have revised the description of those results to reflect the differences. 

3) Examining subgroup differences in attrition is a key valuable component of the study, and to that end the paper would be improved by some examination or at the very least mention of interactive effects between race/ethnicity and gender and attrition. For example, the report Altering the Course: Black Males in Medicine highlights the percentage of black male medical school applicants as the lowest among any subgroup. Would conclusions such as ““Lastly, African American students did not differ from White students in their likelihood of persistence at any point.” (p 19) differ when differentiating between black men and black women?

We think that the reviewer brings up an important issue regarding the interaction between gender and race for likelihood of persistence. We tried to address this issue in two different ways. First, we expanded and restructured Table 1 to describe not only fulfillment rates by gender and by race separately, but also at the intersection of race and gender. This allows us to compare both the number of individuals who declare some intention to pursue medicine as well as persistence patterns over time across demographic groups (e.g., between Black men and Black women). In addition to these descriptives, we also tested the same four models whose results are displayed in Table 3 when including the interaction terms between gender and race dummy variables. Because the inclusion of interaction terms would add an additional table that is similar to but much larger than Table 3 and more cumbersome to interpret, and most of the coefficients are statistically not significant, we did not table those results but describe the significant effects at the end of the Results section. 

Reviewer #4: This study investigates the completion rates of students completing the sequence of undergraduate natural science coursework prerequisite for entry to medical school in the U.S. using a large-scale, multi-institution dataset. It examines the associations of multiple sociodemographic, pre-college achievement, career aspiration, and college-level achievement measures in relation to students’ completion of four milestone events defined by natural science coursework from the first semester through advanced organic or biochemistry courses.

The study showed that, of those who reported intending to pursue a medical degree when they took the SAT in high school, only 39% completed the first in a sequence of courses that together provide the foundation needed to be ready for medical school. Completion rates were progressively higher for the (likely) increasingly select group of students achieving each of the three subsequent milestones of coursework. In the end, the overall completion rate of all four milestones was 16.5%. Observed proportions completing the full sequence were higher for males and Asian students, and lower for females and students from races/ethnicities underrepresented in medicine (i.e., American Indian, Black, and Hispanic).

High school GPA, SAT scores, SES, Gender (Male), and Race (Asian) were associated with greater likelihoods of completing the first set of milestone courses. Gender (Male), Race (Hispanic), and college grades in the first milestone courses were associated with greater likelihoods of completing the second milestone courses, which included the second semester courses of the subjects comprising the first milestone. SAT scores and high school GPA were associated with greater likelihoods of completing the third milestone courses, and race (Hispanic) with a lower likelihood. Finally, gender (Male) and intention to pursue medical school were associated with greater likelihoods of completing the fourth milestone courses, while SAT scores were associated with a lower likelihood of completing the fourth milestone courses.

Although this study has the potential to enhance our understanding of the points at which those interested in the medical profession drop out, the paper fails to make a strong case for it.

1. The authors should consider rewriting the introduction and discussion with a clearer focus and relevant citations, considering the following as suggestions.

• Integrating studies about the factors that contribute to a lack of diversity in STEM or medicine that occur during or before college with the present study, including those referenced in the current about “the leaky pipeline.” References 4, and 11-19 seem germane to the study’s focus.

• Addressing, alongside things like gender norms that might affect persistence, comparable treatment of the lower rates of completion for those from racial/ethnic backgrounds underrepresented in medicine who, more often have lower-quality middle and high school education. Addressing the role that high school preparation might play in students first-semester performance in natural science coursework and beyond would strengthen the paper.

• Removing or clarifying the relevance of literature on burnout/risk of attrition in medical school (references 3, and 5-9), and references 3, 10, and 20, which about attrition in the medical school in the U.K., which differs substantially from medical school in the U.S. The authors also should consider eliminating references about attrition in U.S. medical school (given that 95% of medical students graduate within five years) or make a stronger case for their relevance.

We have revised the introduction and discussion following some of the reviewers’ suggestions, including removing references of studies of medical school in UK, removing review of the literature on attrition in medical school, and adding some language on systemic underrepresentation of ethnic minority groups in medicine. 

2. With a clearer focus, the authors might present their results differently to highlight important findings. For example, Table 1 shows that Black and Hispanic students completed all four milestones at lower rates than White or Asian students. It might be important, given the research questions, to know at which milestone(s) they did not progress, in addition to showing their prevalence at the two end points. Similarly, it was confusing on page 19 to read that African American students did not differ from White students in their likelihood of persistence at any point, even though only 9% of Black students completed all four milestones, compared to 16% of White students. The authors should address how the observed and predicted rates of completion lead to different interpretations.

3. Similar suggestions would improve the discussion of the paper’s findings.

We have expanded and restructured Table 1 and present fulfillment rates by racial subgroups at various milestones, following the reviewer’s suggestions. We agree that by doing so, we gain a more complete understanding of the process of attrition. 

We thank the reviewer for pointing out the confusion in presenting proportion of fulfillment by group and including those subgroup dummy variables in the regression models. For example, the African American group had the overall lowest proportion of overall eligibility based on their coursework. However, this difference seems to go away in the predictive models once the other predictors (e.g., SES, SAT scores, and grades) are controlled. We have revised the language in the Results and Discussion sections to clarify this. 

4. Finally, the conclusions are unfounded and should be rewritten.

5. Other minor suggestions include:

• addressing how the increasingly select sample of students might affect the results of the logistic regression analyses. For example, SES, SAT scores, and high school GPA may be restricted as the sample reduces from more than 13,000 to about 2,500 students.

• Replacing outdated references with more recent research (e.g., 18) and confirming the appropriateness of journals cited (e.g., 9, which is missing the Journal Name “Journal of Unschooling and Alternative Learning”).

• Clarifying the relevance or eliminating the text describing challenges identifying premed majors. The study documented the process for identifying courses prerequisite for medical school (and that are well described on each medical school’s website).

References have been edited and the Conclusion section was rewritten.

---

## [Decision Letter · Decision Letter 1]

24 Nov 2020

The process of attrition in pre-medical studies: A large-scale analysis across 102 schools

PONE-D-20-15864R1

Dear Dr. Zhang,

We’re pleased to inform you that your manuscript has been judged scientifically suitable for publication and will be formally accepted for publication once it meets all outstanding technical requirements.

Kind regards,

Luisa N. Borrell, DDS, PhD

Academic Editor

PLOS ONE

Additional Editor Comments (optional):

You have addressed the reviewers' comments satisfactorily. However, I have some comments for thee tables. Please see attached.

Reviewers' comments:

Reviewer's Responses to Questions

**Comments to the Author**

1. If the authors have adequately addressed your comments raised in a previous round of review and you feel that this manuscript is now acceptable for publication, you may indicate that here to bypass the “Comments to the Author” section, enter your conflict of interest statement in the “Confidential to Editor” section, and submit your "Accept" recommendation.

Reviewer #2: All comments have been addressed

Reviewer #3: All comments have been addressed

Reviewer #4: All comments have been addressed

2. Is the manuscript technically sound, and do the data support the conclusions?

Reviewer #2: Yes

Reviewer #3: Yes

Reviewer #4: Yes

3. Has the statistical analysis been performed appropriately and rigorously? 

Reviewer #2: Yes

Reviewer #3: Yes

Reviewer #4: Yes

4. Have the authors made all data underlying the findings in their manuscript fully available?

Reviewer #2: No

Reviewer #3: (No Response)

Reviewer #4: No

5. Is the manuscript presented in an intelligible fashion and written in standard English?

Reviewer #2: Yes

Reviewer #3: Yes

Reviewer #4: Yes

6. Review Comments to the Author

Reviewer #2: my comments were very well addressed - thank you

if I have a quibble, it's to ensure that the mean of %fulfill across institutions is sample-size weighted, to take small-N into account

Reviewer #3: (No Response)

Reviewer #4: The revision was thorough and thoughtful. Three minor changes might improve the text:

1. On page 16, admissions criteria are described as "harsh". Stringent might be a more neutral term.

2. On page 17, last line of first partial paragraph, states "demographic characteristics such as having family members who were doctors and higher family income have been found to aid with persistence. The term "aid" implies an active role in persistence. Do you mean "are associated with"?

3. Reference 3 is a study about persistence in the U.K., and reference 5 is about burnout and thoughts of dropping out among U.S. medical students. Neither of these studies is about actual attrition in the U.S. On page 18, References 3 and 5 are used to support a statement about attrition rates in U.S. medical schools.

(Given the low acceptance rates into medical schools (7) and the attrition rates in medical schools (3,5), this early change in education track may actually prevent additional personal resources from being wasted in the process of applying to medical programs or institutional resources from being wasted when students drop out of medical programs.)

7. PLOS authors have the option to publish the peer review history of their article (what does this mean?). If published, this will include your full peer review and any attached files.

Reviewer #2: No

Reviewer #3: No

Reviewer #4: No

---

## [Editor Report · Acceptance letter]

10 Dec 2020

PONE-D-20-15864R1 

The process of attrition in pre-medical studies: A large-scale analysis across 102 schools 

Dear Dr. Zhang:

I'm pleased to inform you that your manuscript has been deemed suitable for publication in PLOS ONE. Congratulations! Your manuscript is now with our production department. 

Kind regards, 

on behalf of

Dr. Luisa N. Borrell 

Academic Editor

PLOS ONE